# Smoking Cessation Programs Are Less Effective in Smokers with Low Socioeconomic Status Even When Financial Incentives for Quitting Smoking Are Offered—A Community-Randomized Smoking Cessation Trial in Denmark

**DOI:** 10.3390/ijerph191710879

**Published:** 2022-08-31

**Authors:** Charlotta Pisinger, Cecilie Goltermann Toxværd, Mette Rasmussen

**Affiliations:** 1Center for Clinical Research and Prevention, Bispebjerg and Frederiksberg Hospital, 2000 Frederiksberg, Denmark; 2Department of Public Health, Faculty of Health Sciences, University of Copenhagen, 2200 Copenhagen, Denmark; 3Danish Heart Foundation, 1120 Copenhagen, Denmark; 4Clinical Health Promotion Centre, WHO-CC, The Parker Institute, Bispebjerg & Frederiksberg Hospital, 2000 Frederiksberg, Denmark; 5Clinical Health Promotion Centre, WHO-CC, Department of Health Sciences, Lund University, 223 62 Lund, Sweden

**Keywords:** smoking cessation, socioeconomic status, financial incentives

## Abstract

Financial incentives offered to those who quit smoking have been found effective, also in persons with low socioeconomic status (SES), but no previous study has investigated who benefits most: smokers with low or high SES. In this community-randomized trial (“Richer without smoking”), three Danish municipalities were randomized to reward persons who were abstinent when attending the municipal smoking cessation program (FIMs) and three municipalities were randomized to spend the same amount on smoking cessation campaigns recruiting smokers to the smoking cessation program (CAMs). The municipalities each received approximately USD 16,000. An intention-to-treat approach was used in analyses. In regression analyses adjusted for individual- and municipal-level differences, we found that smokers with high SES living in FIMs had significantly higher proportion of validated long-term successful quitters (OR (95% CI): 2.59 (1.6–4.2)) than high-SES smokers living in CAM. Smokers with low SES, however, did not experience the same benefit of financial incentives as smokers with high SES. Neither the FIMs nor the CAMs succeeded in attracting more smokers with low SES during the intervention year 2018 than the year before. Our study showed that smokers with low SES did not experience the same benefit of financial incentives as smokers with high SES.

## 1. Introduction

Even though smoking prevalence has decreased significantly in many countries, population growth has led to an increase in the total number of smokers worldwide, with tobacco smoking causing almost eight million deaths—including one in five deaths in males worldwide [1].

Article 14 of the WHO Framework Convention on Tobacco Control recommends that all countries provide evidence-based support for smoking cessation, such as behavioral support and smoking cessation pharmacotherapy. However, quit rates are often low and, therefore, new ways to help smokers quit are still sought-after.

Incentives, such as money, might have a strong psychological and motivational power. Experiments have shown that interpersonal rejection and physical pain caused desire for money to increase, while handling money, compared with handling paper, reduced distress over social exclusion and diminished the physical pain of immersion in hot water [2]. A Cochrane review from 2019, based on 33 randomized controlled trials (RCTs), performed in many countries across the world, presented a conclusion of high-certainty evidence that successful quitting in the long term is improved by financial incentives [3].

Our research group tested the use of financial incentives for smoking cessation in a community-randomized trial, “Richer without smoking”, taking part in six Danish municipalities in 2018 [4]. This study showed no difference between financial incentives and professional campaigns regarding recruitment of smokers to a smoking cessation program, but financial incentives had significantly greater effect in terms of helping smokers remain smokefree for a year.

Financial incentives have been found effective in many settings (e.g., primary care, college, rural village) and in many different populations (e.g., persons with mental health problems, cancer patients, pregnant women) [3]. However, no previous study has, to the best of our knowledge, investigated who benefits most from financial incentives: smokers with low or those with high socioeconomic status (SES).

It is well documented that there is a strong social gradient in smoking. The poorer and/or less educated a person is and the more mental health or abuse problems a person has, the higher the prevalence of smoking [5,6,7]. Studies have suggested that the high smoking prevalence may be a result of low tobacco health risk awareness, widespread smoking among family and friends, low social support for quitting, stronger addiction to tobacco, lower likelihood of using pharmacotherapy or completing smoking cessation courses, psychological differences such as lack of self-efficacy, and tobacco industry marketing [7,8]. As a result of this, poor, low-educated persons carry the heaviest burden of smoking [9]. Evidence of interventions that work among lower socioeconomic groups is sparse.

All smokers in Denmark have access to smoking cessation interventions in the local community without referral and free of charge. A national smoking cessation gold standard program was developed and is shown to be highly effective [10], even for disadvantaged smokers [11]. The smoking cessation gold standard program offers repeated counselling, including recommended smoking cessation pharmacotherapy by specially trained counsellors.

The “Richer without smoking” study was performed in a general population, in municipalities offering the smoking cessation gold standard program. Already, from the conceptualization phase of the study, we hypothesized that smokers with low SES would benefit more from financial incentives than smokers with high SES, as the reward would be perceived as being higher among those with low SES and would therefore be more appreciated and more motivating. We have information on length of education and employment status which opens the possibility to explore the effect of financial incentives across SES.

The aim of this paper was to investigate if smokers with low SES included in the “Richer without smoking” study [4] in municipalities who offered financial incentives (FIMs) achieved higher levels of abstinence compared to smokers with low SES in municipalities who used the resources on smoking cessation campaigns (CAMs). The same was investigated for smokers with high SES. Further, we wanted to investigate if FIMs were able to attract more smokers with low SES in their smoking cessation program compared with CAMs.

## 2. Materials and Methods

In this community-randomized trial (Figure 1) called “Rigere uden røg” (Richer without smoking), we randomized six Danish municipalities to be either financial incentives municipalities (FIMs) (three municipalities) or campaign municipalities (CAMs) (three municipalities). To reflect a real-world implementation, the influence of the researchers was kept at a minimum throughout the trial.

Inclusion of the intervention municipalities: The principal investigator invited all 29 municipalities in the Capital Region of Denmark to take part in the research project, and six agreed (Figure 1). The local ethics committee decided that there was no need for approval (Journal-nr.: 18024988). The Danish Data Protection Agency also decided that approval was not needed, as the national Smoking Cessation Database [12] (registered and analyzed data) and the national quit line (performed follow-up on long-term abstinence) already had existing data processor agreements. The ClinicalTrials.Gov ID is NCT03849092. Investigating the outcomes across socioeconomic groups was the original plan of the study. No personal data were available to the researchers at any time during the study.

For details, please see Pisinger et al. (“Are financial incentives more effective than health campaigns to quit smoking? A community-randomized smoking cessation trial in Denmark”) [4].

### 2.1. Participants

In this trial we employed no eligibility criteria as the participants were included through already implemented smoking cessation programs in existing smoking cessation services. All smokers enrolling in a municipal smoking cessation gold standard program were included, but the smoking cessation counsellors aimed to enlist as many low-SES smokers as possible (based on short education and/or unemployment).

### 2.2. Randomization and Blinding

Given the nature of the trial, blinding of neither the researchers nor the statistician was possible. However, the statistician was the only one with access to the data. For details, please see [4].

### 2.3. Interventions in the Randomized Trial

#### 2.3.1. Intervention in Financial Incentives Municipalities (FIMs)

The purpose of the three FIMs was to reward smokers for being smokefree when they attended the smoking cessation gold standard program. To do so, each FIM received DKK 100,000 (approximately USD 16,000). Each ex-smoker could receive a maximum of DKK 1200 (approximately USD 190), given as up to four vouchers for a local shopping mall (with a grocery store). Vouchers worth DKK 200 (approximately USD 32) were issued to ex-smokers at the third, fourth, and fifth session. Being smokefree at the last session (abstinence for a total of 4–6 weeks) triggered a final voucher worth DKK 600 (approximately USD 95) (see Appendix A for a graphical presentation of the intervention). Smokers who did not know about the financial incentives when they signed up for the smoking cessation group also received incentives when being abstinent. Abstinence from smoking was confirmed by carbon monoxide levels *<* 10 ppm [13], measured by the smoking cessation counsellor. The FIMs did not receive any money for recruitment of smokers. The municipalities had different recruitment methods before the study started, and little money ear-marked for recruitment. All advertised for their smoking cessation services on their website and most of them had some collaboration with pharmacies and/or general practitioners. We gave very simple examples of how/where they might recruit but no assistance to recruit smokers. The smoking cessation counsellors advertised on their website, informed general practitioners and pharmacies about the project, and provided them with flyers. Further, they hung up homemade photocopied A4/A3 posters, e.g., in vocational schools or in residential areas with many persons with low SES. The intervention was fully delivered in one in three municipalities only. In one municipality, one of the two smoking cessation counsellors was on sick leave for half of the intervention year. In another municipality, all smoking cessation activities were stopped after approximately eight months due to a local political decision. For details, please see [4].

#### 2.3.2. Intervention in Campaign Municipalities (CAMs)

The purpose of the three CAMs was to run campaigns to attract and encourage smokers with low socioeconomic status to sign up for a municipal smoking cessation gold standard program. To do so, each CAM received DKK 100,000. Several advertising agencies were contacted by the CAMs, and, finally, they chose an agency offering a combination of postcards, big and small posters, and online and social media solutions. The campaign material included photos, e.g., of a dog begging to be taken outside or an elderly couple hugging, and each had a short humorous rhyme. Photos, e.g., of a young car mechanic or ordinary elderly people, that hopefully would arouse recognition and appeal to persons with low SES were chosen. The material incorporated the fact that smokers could increase their chances of becoming smokefree on the order of five times (based on Danish data) when participating in a smoking cessation gold standard program compared to trying themselves. Smokers could send a text to 1231 to be contacted by a smoking cessation counsellor. Banners and posters were placed on buses, at bus stops, at main roads, near malls, etc. Postcards/leaflets were placed in public buildings and selected workplaces, as well as being distributed in deprived areas via private mailboxes. Digital marketing was surfaced, e.g., in buses, on main roads, and on Facebook. Smokers attending a smoking cessation gold standard program in a campaign municipality (CAM) were not rewarded financially for abstinence.

### 2.4. Outcome

The intervention took place throughout 2018. Outcomes were registered in the intervention year (2018), as well as in 2017. The primary outcome regarding recruitment was the number of smokers attending a smoking cessation gold standard program. Successful quitting was measured multiple times. In this paper, we present (1) “Carbon-monoxide-validated continuous abstinence after 4–6 weeks” (at the last smoking cessation gold standard program session), (2) “Self-reported 6-months continuous abstinence (not smoking at all after the quit date)”, and (3) “Carbon-monoxide-validated 12+-months point prevalence (smokefree for at least the last 14 days) in the randomized trial. Six- and 12+-months information was obtained by the national quit line. Participants who reported to be smokefree after 12+-months were contacted by a local smoking cessation counsellor to arrange for a carbon monoxide measurement to verify the abstinence.

### 2.5. Statistics

Results were reported as percentages and absolute numbers, and the χ^2^ test was used to compare baseline characteristics.

The variable *socioeconomic status* was defined by education and employment status. Low SES was defined as having no education except school for up to 12 years, or short work-related courses only *or* as being unemployed (and receiving welfare benefits while being of working age; “housewives” and students were not included in this group) [11].

In the recruitment analyses, the numeric change in number of participants between 2017 and 2018 was calculated for each municipality. The differences were then tested using a linear regression model, adjusted for free/subsidized nicotine replacement therapy ((NRT); not given to all smokers by all municipalities) and number of smokers in 2017, as these diverged significantly across the municipalities.

Successful quitting was presented as raw abstinence rates (proportion of successful quitters at follow-up) stratified by random allocation and SES. In the main analyses of successful quitting, we used the intention-to-treat (ITT) approach assuming all non-respondents to be smokers. Odds ratios (ORs) and 95% confidence intervals (95% CI) were estimated by running a mixed-effect logistic regression analysis using the municipality as the first-level cluster (random effect), while all other predictors were modeled as fixed effects.

Initially, analyses were performed to test for interaction between random allocation and SES, and the following analyses were stratified by random allocation and SES.

First, the univariate analyses were conducted, followed by the fitting of the multivariable model, including predictors based on the initial results and factors known from the literature to be associated with the outcome. Furthermore, we included municipality-level factors, taking the difference between municipalities into account. The fully adjusted model included sex (male/female), age (15–24, 25–34, 35–44, 45–54, 55–66, 67+ (retirement age)), Fagerström Test for Nicotine Dependence [14] (low: 0–6 points/high: 7–10 points), number of cigarettes smoked per day, and the quit rate in 2017. Finally, the proportion of smokers who received free/subsidized nicotine replacement treatment (NRT) was included. All variables were entered together and listwise deletion was employed, excluding smokers with missing values from the analyses. Sensitivity analyses using smokers with valid follow-up only (complete case approach) were performed, using the same adjusted model.

## 3. Results

### 3.1. Attrition and Loss to Follow-Up

In 2018, a total of 295 smokers attended the smoking cessation gold standard program in the financial incentives municipalities (FIMs), and 580 in the campaign municipalities (CAMs). The distribution of smokers with low and high SES was the same in both intervention groups (48% low SES and 52% high SES). Loss to follow-up was higher among smokers with low SES in both the FIMs and the CAMs at all timepoints. After 6 and 12+-months, the attrition was much higher in the CAM groups compared to the FIMs (Figure 1). Loss to follow-up did not differ statistically among the groups at the end of the smoking cessation gold standard program (*p* = 0.057), but after 6 and 12+-months there was a statistically significant difference between the four groups (*p* ≤ 0.001 at both timepoints).

### 3.2. Baseline Characteristics of Smokers

The majority of the participants were 45 to 66 years old, and approximately six in ten were women (Table 1). Roughly three in ten were highly addicted to nicotine, and about half of the smokers had low SES. There was no difference in the share of smokers with low SES between the FIMs and the CAMs. The median number of cigarettes per day was 20 across the groups, except for the high-SES group in the FIMs, where it was 16 cigarettes per day.

### 3.3. Recruitment

Neither the financial incentives municipalities (FIMs) nor the campaign municipalities (CAMs) succeeded in attracting more smokers with low SES during 2018 compared with 2017 (*p* = 0.438 and 0.879, respectively) (Table 2). In the adjusted regression analyses, there was no statistically significant difference in recruitment of smokers with low SES between the FIMs and the CAMs during the intervention year (*p* = 0.147). Sensitivity analyses taking the reduced activity in two of three municipalities in the FIMs into account did not change the conclusion (*p* = 0.869).

### 3.4. Abstinence

Regarding the crude abstinence rates, the highest proportion of successful quitters were seen among participants with high SES in the FIMs, both at the end of the smoking cessation gold standard program, after 6 months, and after 12+-months (Figure 2). In both the FIMs and the CAMs, persons with low SES achieved lower abstinence rates than person with high SES. Abstinence rates among the persons with low SES in the FIMs, however, were almost the same as among persons with high SES in the CAMs.

Successful quitting based on a complete case approach can be seen in Appendix B. The highest crude abstinence rates were seen among participants with high SES in the FIMs.

### 3.5. Adjusted ITT Analyses

Statistically significant interactions were found between the intervention groups (FIMs/CAMs) and SES (high/low) after 6 and 12+-months (*p* = 0.003 and *p* = 0.038, respectively). This interaction was not seen at the end of the smoking cessation intervention (*p* = 0.514).

Based on these interactions, the following regression analyses were stratified according to SES (Table 3).

### 3.6. Complete Case Analyses (Appendix C)

Adjusted and unadjusted analyses resulted in essentially similar results. Additionally, we did not observe any noticeable differences between self-reported continuous abstinence after 6 months and validated successful quitting after 12+-months in either the crude or the fully adjusted analyses.

## 4. Discussion

In this large community-randomized trial, we have previously shown that smokers living in municipalities offering financial incentives to those who quit when attending a municipal smoking cessation program achieved significantly higher long-term abstinence rates than smokers living in municipalities using professional campaigns but not offering financial incentives. In this paper, we show that smokers with high SES living in municipalities offering financial incentives achieved significantly higher long-term abstinence rates than smokers with high SES living in municipalities not offering financial incentives. However, it seems that smokers with low SES did not experience the same benefit of financial incentives as smokers with high SES. Furthermore, neither the financial incentives municipalities (FIMs) nor the campaign municipalities (CAMs) succeeded in attracting more smokers with low SES during the intervention year than the year before.

Despite no difference in intention to quit or number of quit attempts, people with low SES are significantly less successful in their quit attempts than those with higher SES [15]. Some of the reasons for this are strong tobacco addiction, less use of smoking cessation pharmacotherapy, quitting of treatment, early drop-out from smoking cessation programs, reduced social support for quitting and low self-efficacy [7,16,17,18]. Few studies have tested which interventions are effective among smokers with low SES [19] and it is crucial that we search new ways to help those with low SES who want to quit.

In 2019, a Cochrane review identified 33 randomized controlled trials (RCTs) investigating the effect of financial incentives [3]. Since then, several other RCTs have confirmed that incentives are an efficient, cheap, and safe method to improve quit rates [4,20,21,22,23,24,25]. Results from several studies indicate that offering small financial incentives for smoking abstinence are also efficient in disadvantaged persons [26,27,28]. The Cochrane review suggested a favorable benefit of incentives for smoking cessation at longest follow-up in substance users. Although confidence intervals were wide due to small number of studies and participants, the point estimate was consistent with the overall meta-analysis [3]. The same was found in studies offering financial incentives to pregnant women who smoked [3]. It is well known that many women who continue smoking during pregnancy have a low SES. A recent RCT including female smokers who were 250% below the federal poverty level also found a beneficial effect of offering incentives [24]. Another study found that use of financial incentives to engage with tobacco quit line treatment is a cost-effective option to enhance smoking cessation rates for low-income smokers [29].

Therefore, we hypothesized that smokers with low SES would have a high benefit of financial incentives, even higher than smokers with high SES. Interviews (unpublished) with persons attending the smoking cessation gold standard program in the FIMs revealed that those with low income especially really appreciated the vouchers, when confirmed smokefree. Unadjusted analyses (Figure 2) showed quit rates comparable with persons with high SES in the CAMs not offering financial incentives. It was therefore surprising that the adjusted analyses could not confirm that smokers with low SES significantly benefited from financial incentives. One explanation could be that our study was under-powered due to the high drop-out rates. The adjusted OR for persons with low SES living in FIMs were higher at one-year follow-up than in low-SES persons in the CAMs, but not significantly higher. Another explanation could be that our definition of persons with low SES did not exclusively include those who were poor and therefore they did not experience the economic reward as very motivating. In Denmark, unemployed persons receive high financial support from the government for a long period of time, and persons with low education might be rich craftsmen or self-employed entrepreneurs. Potentially, a retired person with high education might have a lower income, even though he/she is defined as having a high SES.

It is interesting that persons with high SES responded so strongly to financial incentives, even though they probably did not need them. We hypothesize that it might have given them a bit of a bad conscience that they received the voucher, even though they were not poor. This bad conscience might have been a motivating factor for resourceful persons to stay abstinent. The finding can be very useful for, e.g., private companies with many high-SES employees, as these probably would have a great benefit of being offered a financial incentive if they quit smoking.

Why do incentives work? Surprisingly, the amount of money seems to be of little importance. The Cochrane review found no significant difference in the effect of smoking cessation between trials paying smaller or larger amounts of rewards [3]. Some studies including smokers with low SES found that financial incentives were associated with a higher use of cessation medication [21,30]; in our study, many smokers were offered free/subsidized NRT by the six municipalities. Further, a couple of studies have investigated whether incentives increase self-efficacy. A large workplace study found that the effect of financial incentives on successful quitting was mediated by a higher self-efficacy [30], but another study found that self-efficacy did not mediate the relation between the incentives and smoking cessation in the overall sample [31]. However, there was significant moderating effect of sex, with a stronger association between incentives and greater self-efficacy among females than males, which in turn influenced the likelihood of smoking cessation [31]. One study suggested that the perceived importance of earning abstinence-contingent incentives early in a quit attempt may indicate whether an individual will be responsive to the magnitude of incentives offered [32]. In conclusion, we do not fully know the exact mediators of the effect of incentives.

Even though we hypothesized a greater effect of the financial incentive intervention in low-SES smokers than in high-SES smokers, it is not a unique finding that smokers with low SES benefit less from smoking cessation interventions. A study found that, in general, smoking cessation interventions in Europe seem to have increased inequalities in smoking [33]. An exemption seems to be the United Kingdom’s National Health Service (NHS) stop smoking services. Efforts were made to advertise smoking cessation in disadvantaged communities and to train volunteers or community workers as advisers, an approach that was found to be an effective strategy in treating disadvantaged smokers [34]. In all areas there was “positive discrimination”, meaning that services were reaching a higher proportion of smokers living in the most disadvantaged areas compared with more affluent areas [35]. It was concluded that the smoking cessation service probably made a modest contribution to reducing inequalities in smoking prevalence [36]. Population-level tobacco control interventions might have the potential to benefit more disadvantaged groups and thereby contribute to reducing health inequalities [37], and high tobacco taxes seem to particularly benefit lower socioeconomic groups [38].

Our study indicates that using financial incentives in the general population might, as other smoking cessation interventions, widen the socioeconomic inequality in smoking. This finding needs to be confirmed by other studies investigating the differential impact of financial incentives across socioeconomic groups. If they confirm our finding, one could argue that financial incentives are offered primarily to smokers with low SES to avoid a widening of socioeconomic inequalities in smoking. Several high-quality interventions, mentioned above, have shown that financial incentives do benefit smokers with low SES. On the other hand, denying smokers with high SES an effective treatment might be seen as discrimination. Financial incentives are not only an evidence-based and highly effective smoking cessation method, but the strategy also seems to be highly cost-effective (a good investment) [39,40,41,42]. Therefore, one could argue that it should be offered to as many smokers as possible. Evidence on smoking cessation interventions that work among lower socioeconomic groups and do not widen the socioeconomic inequality in smoking is needed.

Our study also found that neither the FIMs nor the CAMs succeeded in attracting more smokers with low SES during the intervention year than the year before. Previous studies indicated that financial incentives can be a good recruitment method to smoking cessation activities [43,44], even for disadvantaged smokers [39]. Evidence on interventions that are successful in attracting smokers with low SES to smoking cessation services is needed.

### 4.1. Strengths

It is a strength that this pragmatic RCT, in addition to testing the effect of the intervention, also tested the feasibility of implementing an offer of financial incentives to smokers on a municipal level. It is also a strength that we adjusted for several important confounders both at the municipal and at the individual level. Both intention-to-treat and sensitivity analyses using the complete case approach were performed, and abstinence was validated by carbon monoxide measurements at the end of the smoking cessation gold standard program and after 12+-months.

### 4.2. Weaknesses

It was a challenge to evaluate these community interventions in a real-life setting as it deviates from the framework of an RCT which has strict requirements for evaluation and is conducted within a more rigorously controlled clinical research environment [45]. The high attrition, especially after 12+-months, is considered the most important weakness. The definition of socioeconomic status has weaknesses. We had information on employment and education but not on economy. A person with low education who has had very good income but is unemployed or on sick leave for a shorter period of time would be categorized as having low SES. A person with, e.g., abuse, no education, very bad finances, and no employment during most of his/her life would end up in the same category. Additionally, it must be considered that the study was performed in a high-income country. The amount a participant could earn was relatively small in a Danish context, while the same amount would have a much greater relative value in a low-income country. Therefore, the results are not necessarily transferable to low-income countries. For further considerations on weaknesses in the design of the study, please see the previous paper [4].

## 5. Conclusions

In a large community-randomized trial, we found that smokers with high socioeconomic status (SES) living in municipalities offering financial incentives achieved significantly higher long-term abstinence rates than smokers with high SES living in municipalities not offering financial incentives but instead running campaigns to attract smokers to smoking cessation services. Consequently, smokers with low SES did not experience the same benefit of financial incentives as smokers with high SES. Furthermore, neither the financial incentive municipalities nor the campaign municipalities succeeded in attracting more smokers with low SES during the intervention year than the year before. Evidence on recruitment methods to smoking cessation services and on smoking cessation interventions that work among lower socioeconomic groups is needed.

## Figures and Tables

**Figure 1 ijerph-19-10879-f001:**
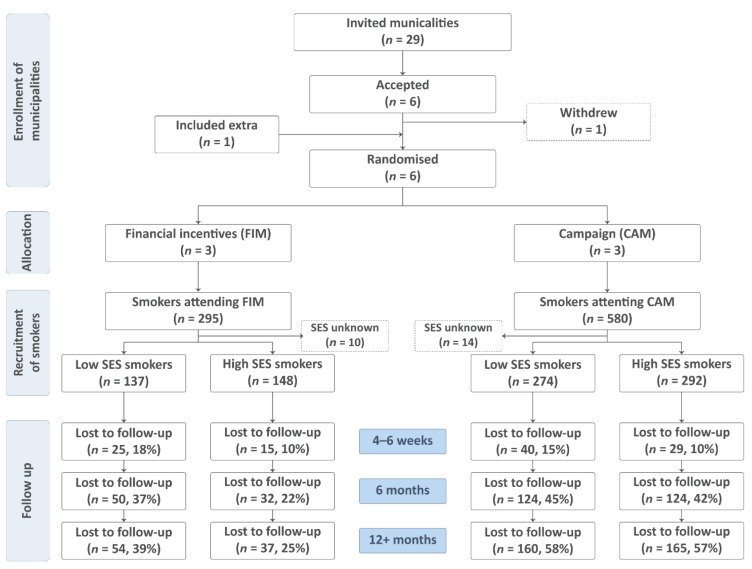
Flowchart of the allocation of municipalities and recruitment and follow-up status of smokers included in the community-randomized trial “Richer without smoking”, Denmark.

**Figure 2 ijerph-19-10879-f002:**
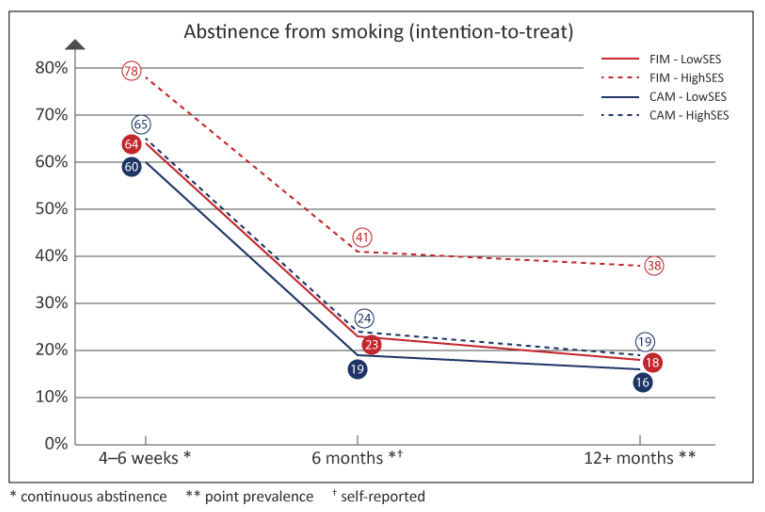
Abstinence rates from smoking in the “Richer without smoking” study, based on the intention-to-treat principle; Denmark, 2018. Validated abstinence rates at the end of the intervention (4–6 weeks), after 12+-months, and self-reported abstinence rates after 6 months. FIM: financial incentive municipality, CAM: campaign municipality, SES: socioeconomic status.

**Table 1 ijerph-19-10879-t001:** Baseline characteristics of smokers with high and low socioeconomic status (SES) included in the community-randomized trial “Richer without smoking”, Denmark.

Characteristics	Financial Incentives Municipalities				Campaign Municipalities			
	Low SES		High SES		Low SES		High SES	
	n	%	n	%	n	%	n	%
Total	137	48.1%	148	51.9%	274	48.4%	292	51.6%
Sex								
Men	57	41.6%	59	39.9%	114	41.6%	121	41.4%
Women	80	58.4%	89	60.1%	160	58.4%	171	58.6%
Age								
Up to 24	8	5.8%			29	10.6%	12	4.1%
25–34	12	8.8%	12	8.1%	35	12.8%	33	11.3%
35–44	21	15.3%	21	14.2%	41	15.0%	41	14.0%
45–54	38	27.7%	30	20.3%	56	20.4%	70	24.0%
55–66	49	35.8%	42	28.4%	90	32.8%	76	26.0%
67+	9	6.6%	43	29.1%	23	8.4%	60	20.5%
Heavy smokers								
No (1–14 cigarettes/day)	21	15.3%	31	20.9%	71	25.9%	88	30.1%
Yes (15+ cigarettes/day)	116	84.7%	117	79.1%	203	74.1%	204	69.9%
Fagerström score ^a^								
Low (0–6)	89	65.0%	106	71.6%	169	61.7%	231	79.1%
High (7–10)	48	35.0%	42	28.4%	105	38.3%	61	20.9%
Free/subsidized NRT ^b^								
No	113	82.5%	99	66.9%	134	48.9%	149	51.0%
Yes	24	17.5%	49	33.1%	140	51.1%	143	49.0%

^a^ Nicotine dependence; ^b^ NRT: nicotine replacement therapy.

**Table 2 ijerph-19-10879-t002:** Recruitment of smokers in the FIMs and CAMs in the year of intervention (2018) and the year before (2017). Data are presented according to socioeconomic status (SES).

	Financial Incentives Municipalities				Campaign Municipalities			
	2017		2018		2017		2018	
	n	%	n	%	n	%	n	%
Socioeconomic status								
Low SES	*84*	44.4%	137	48.1%	146	47.9%	274	48.4%
High SES	*105*	55.6%	148	51.9%	159	52.1%	292	51.6%

**Table 3 ijerph-19-10879-t003:** Unadjusted and adjusted logistic regression analysis of the effect (successful quitting) in the financial incentives municipalities (FIMs) compared with the campaign municipalities (CAMs) at the end of the intervention, and after 6 and 12+-months. The results are based on the intention-to-treat principle and stratified by socioeconomic status (SES). OR: odds ratio, CI: confidence interval.

	Unadjusted Analyses			Adjusted Analyses		
Intervention	OR	[CI 95%]	*p*	OR	[CI 95%]	*p*
Low SES						
CAMs	1.00					
FIMs (4–6 weeks)	1.77	[1.16, 2.70]	0.008 *	1.16	[0.62, 2.18]	0.639
FIMs (6 months)	1.22	[0.74, 2.01]	0.437	0.62	[0.26, 1.45]	0.268
FIMs (12+-months)	1.29	[0.73, 2.28]	0.375	1.17	[0.55, 2.51]	0.677
High SES						
CAMs	1.00					
FIMs (4–6 weeks)	1.84	[1.17, 2.91]	0.009 *	1.51	[0.63, 3.62]	0.344
FIMs (6 months)	2.16	[1.41, 3.30]	≤0.001 *	2.35	[1.21, 4.55]	0.011 *
FIMs (12+-months)	2.69	[1.68, 4.33]	≤0.001 *	2.59	[1.59, 4.22]	≤0.001 *

Adjusted analyses included age, sex, Fagerström score, number of cigarettes smoked daily, abstinence in 2017, proportion of smokers receiving free/subsidized nicotine replacement therapy (NRT), and municipality. * Statistically significant.

## Data Availability

Data used in this study were obtained via an agreement with the national Danish Smoking Cessation Database. These data are not publicly available and therefore cannot be made available according to the regulations of the database.

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
