# Peer review of "Smoking Cessation Programs Are Less Effective in Smokers with Low Socioeconomic Status Even When Financial Incentives for Quitting Smoking Are Offered—A Community-Randomized Smoking Cessation Trial in Denmark"

_ijerph, 2022, doi:10.3390/ijerph191710879_

Round 1

Reviewer 1 Report

Conceptual/Logical structure

It is a highly specialized study performed by a team of experienced authors. The study includes a continuation of the previous research (Bibliography position no. 4).

Although I am not a specialist in such a narrow area yet the relative simplicity of the problem, its popularity in the world, and my competencies in empirical social studies allow me to assess this text. Last but not least, due to various reasons, I know the daily life in Denmark. Obviously, it is not any source of bias but I intend to show my competencies in assessing such a highly specialized text.

The concept of the paper is correctly formulated but some issues have to be touched upon. The aim of the paper is stated very clearly – continuation of the previous research (quotation no. 4). However, the explanation of presenting the study with the result of research done 4 years ago demands a more detailed explanation.

The following questions have to be answered:

1.     To what extent the research presented in the paper constitutes a logical continuation of the previous studies?

I do not criticize this team but it is very common in quantitative empirical social studies that researchers abuse the following situation:

Give me 6,000 numbers (6,000,000 with Big Data) and I will tell you my story. There are always results of the computation and any interpretations are always possible.

2.     As to avoid the above problems the Authors have to show the reasons for the continuation of the previous studies and to show that it is not just further processing of the “numbers” gathered earlier. For example, searching for new phenomena, and references to other similar studies.

It has been already shown (l. 82-87) but perhaps the Authors could find other arguments. It would likely help to increase the cognitive, theoretical, and practical value of the paper. 

There are some examples of “hypothesizing” in the text. Presentation of hypotheses at the beginning of the text could be helpful, and additionally, it would also facilitate answering the questions asked in point 1. The hypotheses defined at the beginning of the paper would be helpful in increasing the logical coherence of reasoning.

Detailed assessment of the paper

Concept

I have no other questions concerning the concept of the paper.

Methods & data

The methods are carefully planned by the experienced authors. No comments on my part. As usual, I trust the validity of the data on the basis of academic integrity. No external availability of the data is declared.

Results

As it is mentioned above, it is a very specialized detailed study developed by experienced team. The discussion and conclusions are correct. As it was also mentioned, that kind of statistical studies ALWAYS gives the results that can be discussed, analyzed, compared, etc. It is not criticism but rather an observation.

It has been omitted in the conclusions that the research has been conducted in a rich country. Conclusions stemming from such a refined project may have limited validity for comparisons with other countries. 

Formal remarks

It would be worthwhile to explain the acronyms either at their first occurrences or perhaps in a separate list, or table.

The text is very carefully edited.

Final opinion

This highly specialist paper can be published with minor suggested changes. 

Author Response

Thank you very much for you valid comments. We have provided a point-to-point letter below: 

Conceptual/Logical structure:

It is a highly specialized study performed by a team of experienced authors. The study includes a continuation of the previous research (Bibliography position no. 4).

Although I am not a specialist in such a narrow area yet the relative simplicity of the problem, its popularity in the world, and my competencies in empirical social studies allow me to assess this text. Last but not least, due to various reasons, I know the daily life in Denmark. Obviously, it is not any source of bias but I intend to show my competencies in assessing such a highly specialized text.

The concept of the paper is correctly formulated but some issues have to be touched upon. The aim of the paper is stated very clearly – continuation of the previous research (quotation no. 4). However, the explanation of presenting the study with the result of research done 4 years ago demands a more detailed explanation.

The following questions have to be answered:

  • To what extent the research presented in the paper constitutes a logical continuation of the previous studies?
    I do not criticize this team but it is very common in quantitative empirical social studies that researchers abuse the following situation:
    Give me 6,000 numbers (6,000,000 with Big Data) and I will tell you my story. There are always results of the computation and any interpretations are always possible.

We can assure the reviewer that it always has been the original plan to investigate the outcome across socioeconomic groups. When we requested the Trygfoundation for funding we actually wrote this as a primary aim, but then we realized that we had to make calculations on the whole population first.

  • As to avoid the above problems the Authors have to show the reasons for the continuation of the previous studies and to show that it is not just further processing of the “numbers” gathered earlier. For example, searching for new phenomena, and references to other similar studies.
    It has been already shown (l. 82-87) but perhaps the Authors could find other arguments. It would likely help to increase the cognitive, theoretical, and practical value of the paper.
    There are some examples of “hypothesizing” in the text. Presentation of hypotheses at the beginning of the text could be helpful, and additionally, it would also facilitate answering the questions asked in point 1. The hypotheses defined at the beginning of the paper would be helpful in increasing the logical coherence of reasoning.

We agree that it is important to make it clear that it is not just further processing of data and that a hypothesis should be presented.  We already wrote in the introduction: “We hypothesised that smokers with low SES would benefit more from financial incentives than smokers with high SES, as the reward would be perceived as being higher among those with low SES and would therefore be more appreciated and more motivating”.

We have now added this: “Already from the conceptualization phase of the study we hypothesised that smokers with low SES would benefit ….”.

Detailed assessment of the paper:

Concept:

  • I have no other questions concerning the concept of the paper.

Methods & data

  • The methods are carefully planned by the experienced authors. No comments on my part. As usual, I trust the validity of the data on the basis of academic integrity. No external availability of the data is declared.

Results

  • As it is mentioned above, it is a very specialized detailed study developed by experienced team. The discussion and conclusions are correct. As it was also mentioned, that kind of statistical studies ALWAYS gives the results that can be discussed, analyzed, compared, etc. It is not criticism but rather an observation.
  • It has been omitted in the conclusions that the research has been conducted in a rich country. Conclusions stemming from such a refined project may have limited validity for comparisons with other countries.

Thank you for the important point. We have added this in the discussion: “Also, it must be considered that the study was performed in a high-income country. The amount a participant could earn was relatively small in a Danish context while the same amount would have a much greater relative value in a low-income country. Therefore, the results are not necessarily transferable to low-income countries”.

Formal remarks

  • It would be worthwhile to explain the acronyms either at their first occurrences or perhaps in a separate list, or table.

We agree that it is frustrating with many acronyms. We have now added a box with acronyms below figure 1 and removed the acronyms SC, CO and SC-GSP. All acronyms are explained first time they are mentioned. The acronyms FIM and CAM are explained on each page. We hope it makes the paper easier to read.

  • The text is very carefully edited.

Final opinion

  • This highly specialist paper can be published with minor suggested changes

Reviewer 2 Report

The manuscript consists of total 13 pages, including 3 tables, 2 figures and the list of total 42 literature references. The original material-based article presents the results of the study concerning the efficacy of public health intervention on the common modifiable serious health risk factor - smoking - through financial incentives. As such, the topic fits into the scope of the works published by the Journal.

The title of the manuscript is inviting and intriguing, but it is not fully relevant to the contents of the article - actually it shall be e.c. "Smoking cessation programs are less effective in smokers with low socioeconomic class even when financial incentives for quitting smoking are offered." The main investigated problem here is not that the low socioeconomic class smokers get less money than the high socioeconomic class smokers for participation in the smoking quitting programs - because it is a natural and logical consequence if (as the study results demonstrated) the former drop out more often and earlier than the latter, even if they are offered money for participation (which is the actual problem to be stressed here). Drawing conclusions stressing the aspect of the findings that introducing financial incentives into the smoking programs (from which the low socioeconomic class smokers drop out more often even though the given sum of money is comparably more important to them than to the high socioeconomic class smokers) would increase any socioeconomic inequality in the society is peculiar - the actual finding is that the Authors identified a means of persuasion that targets satisfactorily the high socioeconomic class smokers (which is a success) and some different tools need to be sought for further for the application with the lower socioeconomic class smokers; presenting a success as a threat merely just because it is not equally applicable to all target groups is in my opinion a mistake. Therefore, in my opinion again, the Authors shall also consider rewriting the second sentence of the Conclusions e.c. "CONSEQUENTLY, smokers with low SES did not experience the same benefit of financial incentives as smokers with high SES." and removing the fourth sentence of Conclusions at all.

Although the text is written in good quality English, it is rather difficult to read and follow for the Reader as the Authors tend to use quite long and composed sentences, often in combination with long unique names and/or multiple abbreviations, typically explained only in the very beginning of the quite long text (there are many of them and this may make the Reader confused and frustrated as they need to be looked up again); for the sake of clarity, it would be advised to use more but shorter sentences instead and once per page add the full name together with the abbreviation if it is used again.

The Abstract mirrors the crucial contents of the main article; the "SC" abbreviation needs to be explained while used for the first time.

The Introduction section provides enough insight into the background of the project.

The Material and methods section is detailed but each of the actual interventions would be easier to grasp for the Readers if presented in some graphical form so the core actions are clear at first sight: who got what for doing what and for how many times. The Authors shall rather refer to the name of the published work author by his/her name instead of referring incompletely "For details, please see [4]."

The Results are consistent with the described methodology and presented in enough detail.

The Discussion places the Authors own results in the context of previously published works but the Authors shall take into account the possibility of altering the direction of the line of reasoning considering the remarks made above.

The Conclusions shall be changed, considering the remarks made above. In my opinion, in Conclusions all abbreviations shall be accompanied with respective full names so the communicate is maximally clear to the Readers who would jump directly to this section as many of us do while beginning reading scientific papers.

The figures with captions shall provide the possibility to be interpreted independently of the text so the abbreviations used shall be explained in the captions so the Readers do not need to look up for explanations of abbreviated terms in the main text.

The literature references are reasonably recent and relevant to the topic of the article.

Author Response

Thank you very much for you valid comments. We have provided a point-to-point letter below:

The manuscript consists of total 13 pages, including 3 tables, 2 figures and the list of total 42 literature references. The original material-based article presents the results of the study concerning the efficacy of public health intervention on the common modifiable serious health risk factor - smoking - through financial incentives. As such, the topic fits into the scope of the works published by the Journal.

  • The title of the manuscript is inviting and intriguing, but it is not fully relevant to the contents of the article - actually it shall be e.c. "Smoking cessation programs are less effective in smokers with low socioeconomic class even when financial incentives for quitting smoking are offered." The main investigated problem here is not that the low socioeconomic class smokers get less money than the high socioeconomic class smokers for participation in the smoking quitting programs - because it is a natural and logical consequence if (as the study results demonstrated) the former drop out more often and earlier than the latter, even if they are offered money for participation (which is the actual problem to be stressed here). Drawing conclusions stressing the aspect of the findings that introducing financial incentives into the smoking programs (from which the low socioeconomic class smokers drop out more often even though the given sum of money is comparably more important to them than to the high socioeconomic class smokers) would increase any socioeconomic inequality in the society is peculiar - the actual finding is that the Authors identified a means of persuasion that targets satisfactorily the high socioeconomic class smokers (which is a success) and some different tools need to be sought for further for the application with the lower socioeconomic class smokers; presenting a success as a threat merely just because it is not equally applicable to all target groups is in my opinion a mistake. Therefore, in my opinion again, the Authors shall also consider rewriting the second sentence of the Conclusions e.c. "CONSEQUENTLY, smokers with low SES did not experience the same benefit of financial incentives as smokers with high SES." and removing the fourth sentence of Conclusions at all.

We have changed the title, removed fourth sentence and rewritten second sentence as recommended by reviewer.

  • Although the text is written in good quality English, it is rather difficult to read and follow for the Reader as the Authors tend to use quite long and composed sentences, often in combination with long unique names and/or multiple abbreviations, typically explained only in the very beginning of the quite long text (there are many of them and this may make the Reader confused and frustrated as they need to be looked up again); for the sake of clarity, it would be advised to use more but shorter sentences instead and once per page add the full name together with the abbreviation if it is used again.

We agree that it is frustrating with many acronyms. We have now added a box with 5 acronyms below figure 1 and removed the acronyms SC, CO and SC-GSP (written in full). All acronyms are explained first time they are mentioned. The acronyms FIM and CAM are explained on each page. We have also split long sentences in two, whenever meaningful. We hope it makes the paper easier to read.

Abstract

  • The Abstract mirrors the crucial contents of the main article; the "SC" abbreviation needs to be explained while used for the first time.

Thank you so much for pointing this out. We have now replaced “SC” with “smoking cessation”.

Introduction

  • The Introduction section provides enough insight into the background of the project

Material and methods

  • The Material and methods section is detailed but each of the actual interventions would be easier to grasp for the Readers if presented in some graphical form so the core actions are clear at first sight: who got what for doing what and for how many times. The Authors shall rather refer to the name of the published work author by his/her name instead of referring incompletely "For details, please see [4]."

We have added a simple illustration of the smoking cessation intervention used in the municipalities and the timing and amount of rewards given in the financial incentives municipalities as appendix A.
We have now refered to the name of the published work (reference 4) as suggested.

Results

  • The Results are consistent with the described methodology and presented in enough detail

Discussion

  • The Discussion places the Authors own results in the context of previously published works but the Authors shall take into account the possibility of altering the direction of the line of reasoning considering the remarks made above.

We are not completely sure precisely what reviewer wants us to change. We still find the “flow” and line of reasoning meaningful. We also still believe that it is important to discuss if financial incentive could have an adverse effect (widening the social inequality) if they were used as a population based strategy. As those with high SES benefit more than those with low SES.

We have rewritten a part in the discussion: “Our study indicates that using financial incentives in the general population might, as other smoking cessation interventions, widen the socioeconomic inequality in smoking. This finding needs to be confirmed by other studies investigating the differential impact of financial incentives across socioeconomic groups. If they confirm our finding one could argue that financial incentives are offered primarily to smokers with low SES to avoid a widening of socioeconomic inequalities in smoking. Several high-quality interventions, mentioned above, have shown that financial incentives do benefit smokers with low SES. On the other hand, denying smokers with high SES an effective treatment might be seen as discrimination. Financial incentives are not only an evidence based and highly effective SC method, but the strategy also seems to be highly cost-effective (a good investment) [39-42]. Therefore, one could argue that it should be offered to as many smokers as possible.”

We have also added this sentence: “The finding can be very useful for e.g., private companies with many high SES employees, as these probably would have a great benefit of being offered a financial incentive if they quit smoking”.

We hope that this is acceptable.

Conclusions

  • The Conclusions shall be changed, considering the remarks made above. In my opinion, in Conclusions all abbreviations shall be accompanied with respective full names so the communicate is maximally clear to the Readers who would jump directly to this section as many of us do while beginning reading scientific papers.

We have changed the conclusions, also in the abstract. In both the conclusion in main text (only SES remains) and in the whole abstract we now write full names before all abbreviations.

Figures

  • The figures with captions shall provide the possibility to be interpreted independently of the text so the abbreviations used shall be explained in the captions so the Readers do not need to look up for explanations of abbreviated terms in the main text.

Thank you noticing. We have made the appropriate changes in figure 1, table 1, table 2, figure 2, table 3, Appendix B, and Appendix C.

References

  • The literature references are reasonably recent and relevant to the topic of the article